# Pancreatic Ductal Adenocarcinoma: Epidemiology and Risk Factors

**DOI:** 10.3390/diagnostics11030562

**Published:** 2021-03-20

**Authors:** Jun Ushio, Atsushi Kanno, Eriko Ikeda, Kozue Ando, Hiroki Nagai, Tetsurou Miwata, Yuki Kawasaki, Yamato Tada, Kensuke Yokoyama, Norikatsu Numao, Kiichi Tamada, Alan Kawarai Lefor, Hironori Yamamoto

**Affiliations:** 1Department of Medicine, Division of Gastroenterology, Jichi Medical University, 3311-1 Yakushiji, Shimotsuke, Tochigi 329-0498, Japan; j.ushio@jichi.ac.jp (J.U.); r1403ie@jichi.ac.jp (E.I.); kozue_ando@jichi.ac.jp (K.A.); m05069hn@jichi.ac.jp (H.N.); tetsurou_miwata@yahoo.co.jp (T.M.); kawasakiyuki1219@gmail.com (Y.K.); tadayamatoday@gmail.com (Y.T.); r0760ky@jichi.ac.jp (K.Y.); numawo@jichi.ac.jp (N.N.); tamadaki@jichi.ac.jp (K.T.); yamamoto@jichi.ac.jp (H.Y.); 2Department of Surgery, Jichi Medical University, 3311-1 Yakushiji, Shimotsuke, Tochigi 329-0498, Japan; alefor@jichi.ac.jp

**Keywords:** pancreatic cancer, pancreatic ductal cell carcinoma, epidemiology, risk factor

## Abstract

The number of new cases of pancreatic ductal adenocarcinoma is increasing with a cumulative total of 495,773 cases worldwide, making it the fourteenth most common malignancy. However, it accounts for 466,003 deaths per year and is the seventh leading cause of cancer deaths. Regional differences in the number of patients with pancreatic ductal adenocarcinoma appear to reflect differences in medical care, as well as racial differences. Compared to the prevalence of other organ cancers in Japan, pancreatic ductal adenocarcinoma ranks seventh based on the number of patients, eighth based on morbidity, and fourth based on the number of deaths, with a continuing increase in the mortality rate. Risk factors for developing pancreatic ductal adenocarcinoma include family history, genetic disorders, diabetes, chronic pancreatitis, and intraductal papillary mucinous neoplasms. An issue that hinders improvement in the prognosis of patients with pancreatic ductal adenocarcinoma is the development of a strategy to identify patients with these risk factors to facilitate detection of the disease at a stage when intervention will improve survival.

## 1. Introduction

The number of patients with pancreatic ductal adenocarcinoma (PDAC) continues to increase globally. Although the mortality rates of patients with gastric, colorectal, and hepatic cancers are decreasing because of advances in treatment, there has been no sign of a decrease in the mortality of patients with PDAC. There have been no concrete measures for improving the prognosis of patients with PDAC. Reducing the number of deaths due to PDAC is an urgent issue in global public health. This article describes the epidemiology of PDAC to provide an understanding of the current status of PDAC in the world and discusses the risk factors for the development of PDAC to understand and identify the groups at risk of developing it.

## 2. Epidemiology

### 2.1. Status in the World

To compare the prevalence of PDAC and malignancies in other organs, it is important to consider the number of patients with PDAC, morbidity rate, number of deaths, and mortality rate. In particular, comparisons of global cancer surveys often reflect the state of healthcare in each country. According to a survey by GLOBOCAN in 2020, the number of new cases of PDAC was 495,773, placing it fourteenth on the list of most common cancers. However, the number of deaths secondary to PDAC was 466,003 per year, making it the seventh leading cause of cancer deaths [1]. The one-year overall survival (OS) rate of patients with PDAC is very low (24%), and the five-year OS rate is even lower (9%) [2]. The incidence of PDAC varies among countries, with the highest age-standardized incidence rates in Europe and North America, and the lowest rates in Africa and Central Asia [1]. The incidence of PDAC is slightly higher in males (5.7 per 100,000, with 262,865 cases) than in females (4.1 per 100,000, with 232,908 cases) [1]. Among males, the incidence is high in Eastern (9.3 per 100,000) and Western Europe (9.8 per 100,000)—particularly Hungary (13.7 per 100,000), Latvia (11.9 per 100,000), and Estonia (11.5 per 100,000)—followed by North America (9.9 per 100,000) and Southern Europe (8.4 per 100,000). The incidence is low in West (2.2 per 100,000), Middle (2.0 per 100,000), and East Africa (2.0 per 100,000), as well as in South Central Asia (1.5 per 100,000), particularly Guinea (1.3 per 100,000), India (1.2 per 100,000), and Malawi (0.46 per 100,000) [1] (Figure 1). Comparatively, among females, the incidence of PDAC is high in Western Europe (7.4 per 100,000), North America (6.7 per 100,000), Northern Europe (6.7 per 100,000), and Australia/New Zealand (6.7 per 100,000), particularly Hungary (9.2 per 100,000), the United States of America (7.0 per 100,000), and Sweden (7.9 per 100,000). It is low in West (1.8 per 100,000), East (1.7 per 100,000), and Middle Africa (1.2 per 100,000), as well as in South Central Asia (0.9 per 100,000), particularly Guinea (0.78 per 100,000), India (0.69 per 100,000), and Malawi (0.46 per 100,000) [1]. Wong et al. reported a higher incidence of PDAC in developed countries than in developing countries [3]. Saad et al. reported an annual increase of 1.03% in age-adjusted PDAC incidence according to a report using the Surveillance, Epidemiology, and End Results Program in the United States [4]. If PDAC continues to increase at this rate, it is expected to become the second leading cause of cancer death in the United States by 2030 [5].

The mortality rate associated with PDAC also varies among countries. Europe has the highest PDAC mortality rate (7.2 deaths per 100,000 people), followed by North America (6.5 per 100,000 people). East Africa has the lowest PDAC mortality rate (1.2 per 100,000 people). At least 90% of PDAC deaths are reported in patients aged ≥ 55 years, indicating increased onset of the disease with increasing age. When PDAC mortality was compared by country, mortality rates in males and females were found to be highest in Hungary (10.3 in males, 6.4 in females) and Uruguay (9.6 in males, 6.3 in females) (Figure 1). Comparatively, mortality rates in males were lowest in Botswana (0.07 per 100,000 people) and Malawi (0.47 per 100,000 people), whereas those in females were lowest in Botswana (0.29 per 100,000 people) and Pakistan (0.29 per 100,000 people) (data not shown). Racial differences in various risk factors and advances in diagnostic techniques and medical devices are assumed to be the reasons for regional differences in the morbidity and mortality of patients with PDAC. Mortality rates of patients with PDAC in low or medium human development index countries are about 4–5 times higher than those in high human development index countries [6]. PDAC treatment thus needs to be internationally standardized.

### 2.2. Current Status in Japan

#### 2.2.1. Number and Incidence of Pancreatic Ductal Adenocarcinoma Cases

Japan has an original system for registering cancer cases. The incidence and mortality of PDAC in Japan have been increasing recently. Based on data in the Japanese cancer registry in 2017, the number of males with cancer was as follows: prostate cancer: 91,215, gastric cancer: 89,331, colorectal cancer: 87,019 (colon cancer: 54,358, rectal cancer: 32,661), lung cancer: 82,880, hepatic cancer: 26,570, PDAC: 21,200, and esophageal cancer: 21,145, with PDAC ranking sixth [7].

Comparatively, the number of females with cancer was as follows: breast cancer: 91,605, colorectal cancer: 66,170 (colon cancer: 47,593, rectal cancer: 18,577), lung cancer: 41,630, gastric cancer: 40,144, uterine cancer: 27,736 (uterine cervix cancer: 11,012, corpus uteri cancer: 16,724), and PDAC: 19,780, with PDAC ranking sixth [7].

The combined number of males and females with cancer included: colorectal cancer: 153,193 (colon cancer: 101,952, rectal cancer: 51,241), gastric cancer: 129,476, lung cancer: 124,510, breast cancer: 92,254, prostate cancer: 91,215, PDAC: 40,981, and hepatic cancer: 39,401, with PDAC ranking sixth [7] (Figure 2).

The incidence of cancer per 100,000 males was: prostate cancer: 147.9, gastric cancer: 144.9, colorectal cancer: 141.2 (colon cancer: 88.2, rectal cancer: 53.0), lung cancer: 134.4, hepatic cancer: 43.1, PDAC: 34.4, and esophageal cancer: 34.3, with PDAC ranking sixth [7].

The incidence of cancer per 100,000 females was: breast cancer: 140.8, colorectal cancer: 101.8 (colon cancer: 73.2, rectal cancer: 28.6), lung cancer: 64.0, gastric cancer: 61.7, uterine cancer: 42.6 (uterine cervix cancer: 16.9, corpus uteri cancer: 25.7), and PDAC: 30.4, with PDAC ranking sixth [7].

The overall incidence of cancer per 100,000 males and females was: prostate cancer: 147.9, colorectal cancer: 120.9 (colon cancer: 80.5, rectal cancer: 40.4), gastric cancer: 102.2, lung cancer: 98.3, breast cancer: 72.8, uterine cancer: 42.6 (uterine cervix cancer: 16.9, corpus uteri cancer: 25.7), PDAC: 32.3, and hepatic cancer: 31.1 [1], with PDAC ranking seventh [7] (Figure 3).

#### 2.2.2. Number of Deaths and Mortality Rate of Patients with Pancreatic Ductal Adenocarcinoma

Based on data from the 2019 Japanese cancer registry, the number of cancer deaths in males was: lung cancer: 53,338, gastric cancer: 28,043, colorectal cancer: 27,026 (colon cancer: 17,517, rectal cancer: 9899), PDAC: 18,124, and hepatic cancer: 16,750, with PDAC ranking fourth [7].

The number of cancer deaths in females was: colorectal cancer: 24,004 (colon cancer: 18,082, rectal cancer: 5922), lung cancer: 22,056, PDAC: 18,232, gastric cancer: 14,888, and breast cancer: 14,839, with PDAC ranking third [7].

The combined number of deaths in males and females was: lung cancer: 75,394, colorectal cancer: 51,420 (colon cancer: 35,599, rectal cancer: 15,821), gastric cancer: 42,931, and PDAC: 36,356, with PDAC ranking fourth [7] (Figure 4).

Cancer mortality rates in males were: lung cancer: 88.6/100,000, gastric cancer: 46.6, colorectal cancer: 45.5 (colon cancer: 29.1, rectal cancer: 16.4), PDAC: 30.1, and hepatic cancer: 27.8, with PDAC ranking fourth [7] (Figure 5).

Cancer mortality rates in females were: colorectal cancer: 37.8 (colon cancer: 28.5, rectal cancer: 9.3), lung cancer: 34.7, PDAC: 28.7, and gastric cancer: 23.4, with PDAC ranking third [7] (Figure 5).

The overall combined cancer mortality rates in males and females were: lung cancer: 60.9, colorectal cancer: 41.6 (colon cancer: 28.8, rectal cancer: 12.8), gastric cancer: 34.7, and PDAC: 29.4, with PDAC ranking fourth [7] (Figure 5).

The five-year OS rate for patients with PDAC was 10.2%, whereas the 10-year OS rate was 6.2%. These prognoses are the poorest when compared with prognoses for malignancies in other organs [8].

## 3. Risk Factors

Early diagnosis is important to improve the prognosis of patients with PDAC. To achieve early diagnosis of patients with PDAC, it is extremely important to treat patients based on a solid understanding of associated risk factors. A variety of risk factors have been reported for the development of PDAC. The major factors can be broadly classified into family history, genetic disorders, complications, and preferences. Since PDAC does not present with specific symptoms, it is difficult to use clinical symptoms for early detection. Patients with risk factors for developing PDAC should be followed with blood tests, imaging techniques, and other screening methods.

It is rare for PDAC to be diagnosed at an early stage, with only about 2% of all PDAC being diagnosed at stages 0 and I [9]. However, according to a report compiled by the Pancreatic Cancer Registry of the Japan Pancreas Society, the five-year OS rates for patients with stage 0 disease was 85.8%, stage Ia—68.7%, and stage Ib—59.7%, which are greater than the OS rate at stage II or higher [10]. This demonstrates that to improve the prognosis of patients with PDAC, a system needs to be in place to efficiently identify patients who have risk factors associated with the disease and facilitate early diagnosis [9,10,11,12,13].

### 3.1. Family History

The proportion of patients with PDAC in their family history is reported to be 3–10% in Japan [14,15]. Familial pancreatic cancer is defined as the presence of two or more first-degree relatives with PDAC, such as a parent, sibling, or child. The risk ratio of developing PDAC is reportedly 6.79 times greater in families with a member who has PDAC. Additionally, if there are patients younger than 50 years of age with PDAC in the same family, the risk ratio increases to 9.31-fold [16]. Thus, it is critical to evaluate the family history of patients with PDAC in clinical practice. The disease has been associated with genetic mutations, including those in *BRCA1/2*, *PALB2*, *CDKN2A*, *LKB1/STK11*, *PRSS1*, and *ATM* [17,18,19,20,21,22,23,24,25]. In the United States and Europe, a registration system for familial pancreatic cancer has been established, and a prospective study and genetic analyses are being conducted. In Japan, the Japan Pancreas Society and the Pancreatic Cancer Action Network Japan (PanCAN, Sodegaura, Japan) have established a familial pancreatic cancer registry (http://jfpcr.com, accessed on 11 February 2021). In the future, this is expected to contribute to the identification and early diagnosis of patients with PDAC.

### 3.2. Genetic Disorders

Hereditary pancreatitis is a genetic condition characterized by recurrent episodes of acute pancreatitis starting in infancy. It is associated with mutations in the *PRSS1* and *SPINK1* genes. The risk of onset of PDAC in patients with hereditary pancreatitis reportedly increases by 60–87-fold with aging [26]. Hereditary breast and ovarian cancer syndrome is associated with germline mutations in *BRCA1/2*, leading to the development of breast and ovarian cancers that run in families [27,28,29]. Patients with hereditary breast and ovarian cancer syndrome also have a 3.5–10-fold increased risk of developing PDAC [17,30,31,32,33,34]. Additionally, the risk of developing PDAC is reportedly high in those with hereditary diseases, such as hereditary nonpolyposis colorectal cancer (Lynch syndrome) [35,36], familial adenomatous polyposis [22,36], Peutz–Jeghers syndrome [20,24,37], and familial atypical multiple mole melanoma syndrome [38]. Although the number of patients with these genetic disorders in Japan is low, they need to be considered in clinical practice (Table 1).

### 3.3. Diabetes Mellitus

The risk of developing PDAC is increased in patients with diabetes mellitus. The risk of onset in patients diagnosed with diabetes mellitus within one year is as high as 5.38-fold greater [39]. Thus, close examination of the pancreas is recommended for patients with new-onset diabetes mellitus or sudden poor glycemic control. In meta-analyses, the risk of PDAC in patients with type 2 diabetes mellitus was as high as 1.94-fold. The risk was 5.38-fold within one year, 1.95-fold within 1–4 years, 1.49-fold within 5–9 years, and 1.47-fold for ≥10 years from the onset of diabetes mellitus, showing PDAC presenting particularly after new-onset or rapid aggravation [39]. Smoking and chronic pancreatitis increase the risk of developing PDAC in patients with diabetes mellitus. No consensus has been reached regarding the effect of antidiabetic medications, and no significant effect was observed in the meta-analyses [40].

### 3.4. Intraductal Papillary Mucinous Neoplasms (IPMNs)

IPMN is characterized by ductal dilatation caused by the proliferation of a papillary epithelial neoplasm capable of producing mucus. An IPMN can evolve by itself into invasive cancer; however, primary PDAC can develop in the pancreas secondary to reasons other than IPMN [41]. The incidence of PDAC in branch-type IPMN is 2%–10%, while the mortality rate of PDAC in branch-type IPMN is 15.8–26-fold higher. In particular, this is reported to be 16.7-fold higher in patients aged ≥ 70 years and 22.5-fold higher in females [42,43].

In a retrospective study of 577 patients with branch-type IPMN who were followed up for a long period in the United States, malignant transformation was observed in 8% of patients within 10 years. The incidence of malignant transformation within five years was 4.3%, of which the incidence of invasive cancer was 2.4% [44]. The incidence of invasive cancer in patients with branch-type IPMN without high-risk stigmata or worrisome features is as high as 18.8-fold higher than that in the general population [44]. Some patients develop unresectable advanced PDAC even if they are followed up periodically at time points recommended by the International IPMN/mucinous cystic neoplasm clinical practice guidelines. Thus, there is a need to identify other factors that predict malignant transformation with high accuracy. A large prospective observational study conducted mainly by the Japan Pancreas Society is ongoing. More data regarding the incidence of PDAC in relation to IPMN are anticipated.

### 3.5. Chronic Pancreatitis

Chronic pancreatitis has been reported to be a risk factor for developing PDAC. In a meta-analysis by Malka et al., the risk of PDAC in patients with chronic pancreatitis was 13.3-fold greater [45]. According to a Danish cohort study of 11,972 patients with chronic pancreatitis in 2014, the hazard ratio for PDAC was 6.9-fold higher: 14.6-fold within 2–4 years and 4.8-fold between 5 and 16 years after being registered as having chronic pancreatitis [46]. The high hazard ratio for PDAC within two years of registration could be because patients with PDAC could have been included with those diagnosed with chronic pancreatitis. In a retrospective study of Japanese patients with chronic pancreatitis, excluding patients diagnosed within two years after diagnosis of chronic pancreatitis, the standardized incidence ratio of PDAC in patients with chronic pancreatitis (with a median follow-up of 5.6 years) was found to be 11.8-fold higher [47]. These results support abstinence from alcohol as an important factor in the prevention of PDAC in patients with chronic pancreatitis.

A new mechanistic definition of chronic pancreatitis was proposed for future studies based on a consensus of the definition, diagnostic criteria, classification, treatment, etc., of the disease [48]. In this definition, a conceptual model with a mechanistic definition proposed that acute pancreatitis and recurrent acute pancreatitis represent one stage on a path to chronic pancreatitis. There may be some cases with a sequential relationship between acute pancreatitis and chronic pancreatitis. A meta-analysis revealed a correlation between acute pancreatitis and the development of PDAC [49]. Patients with a past history of acute pancreatitis require especially careful evaluations from the perspective of pancreatic carcinogenesis.

### 3.6. Obesity

According to the results of a meta-analysis, an increase in body mass index (BMI) of 5 kg/m^2^ leads to a 1.10-fold higher increase in the risk of onset of PDAC, an increase in waist circumference of 10 cm leads to a 1.11-fold higher increase, and an increase in waist-to-hip ratio of 0.1 units leads to a 1.19-fold higher increased risk [50]. Additionally, the risk of PDAC increases 1.49-fold in males with BMI ≥ 35 kg/m^2^ and 2.76-fold in females with BMI ≥ 40 kg/m^2^ [51]. In cohort studies, obesity has also been identified as a risk factor for PDAC for those in their 20s. According to the results of a large-scale cohort study in Japan, in males with BMI ≥ 30 kg/m^2^ in their 20s, the risk of developing PDAC was 3.5-fold higher [52]. In other methods and studies, a BMI ≥ 30 kg/m^2^ increased the risk of onset of PDAC by 1.71-fold compared to BMIs 23–24.9 kg/m^2^ in males [53]. Taking into consideration conditions such as diabetes, there may be a need to improve modern eating habits.

### 3.7. Smoking

Cigarette smoking has been identified as a risk factor for the development of cancer in all organs. Smoking increases the risk of developing PDAC by 1.68-fold, [54] demonstrating an increased risk compared to that of nonsmokers. Moreover, cigarette smoking synergistically increases the risk of developing PDAC in patients with other risk factors [55,56]; thus, patients should be given an explanation on the risk of onset of arteriosclerosis and be advised to stop smoking.

### 3.8. Alcohol Consumption

The risk of developing PDAC reportedly increases 1.22-fold in heavy drinkers (ethanol equivalent of ≥37.5 g/day) [57], and significantly increases in moderate drinkers or those who drink less than moderate amounts. It is unknown whether the increase in risk of onset of PDAC is a direct effect of alcohol consumption or progression secondary to chronic pancreatitis.

### 3.9. Other

The number of patients with PDAC reportedly increases with occupational exposure to chemical substances (chlorinated hydrocarbon) [58], blood type (non-O) [59,60,61] history of Helicobacter pylori infection or gastric ulcer [62,63,64], and hepatitis B virus infection [65]. According to a recent study, the risk of PDAC increases 1.74-fold in patients with periodontal disease/periodontitis. Several dietary factors have been reported as risk factors for the development of PDAC. Consumption of red meat may increase the risk of developing PDAC due to the effect of carcinogens in red meat such as nitrates or N-nitroso compounds [66]. Foods with high glycemic indices, such as potatoes, have been reported as a risk factor for developing PDAC [67,68]. We summarized the risk factors in Table 2.

## 4. Conclusions

The epidemiology and risk factors for PDAC were summarized. The issues identified were that the incidence of PDAC is on the rise, and major regional differences exist globally. Differences in risk factors between races are assumed. Further, there are differences in the provision of medical care among regions. Determining risk factors for PDAC may enable early detection of PDAC by conducting periodic examinations in patients who are especially at risk. We anticipate that gaining a greater understanding of the overall picture of PDAC through epidemiology and risk factors will contribute to improving the prognosis of patients with this disease.

## Figures and Tables

**Figure 1 diagnostics-11-00562-f001:**
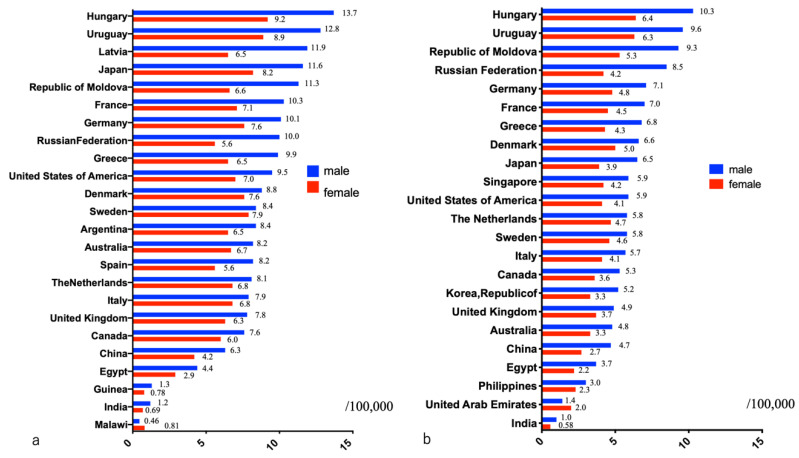
Estimated incidence and mortality from pancreatic ductal adenocarcinoma in 2020 in the world (source: GLOBOCAN 2020 ref. [2]). (**a**) incidence of pancreatic ductal adenocarcinoma; (**b**) mortality rate of patients with pancreatic ductal adenocarcinoma.

**Figure 2 diagnostics-11-00562-f002:**
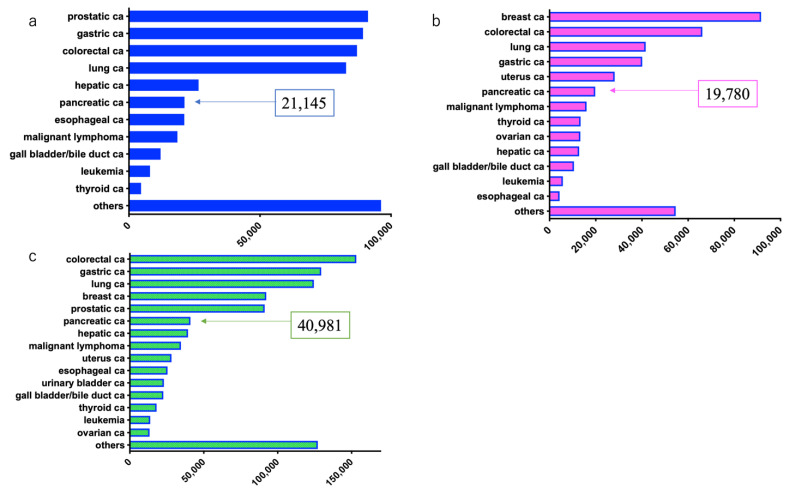
Estimated number of new cases of pancreatic ductal adenocarcinoma in 2017 in Japan (source: Cancer Registry and Statistics ref. [7]). (**a**) Males; (**b**) Females; (**c**) Both males and females.

**Figure 3 diagnostics-11-00562-f003:**
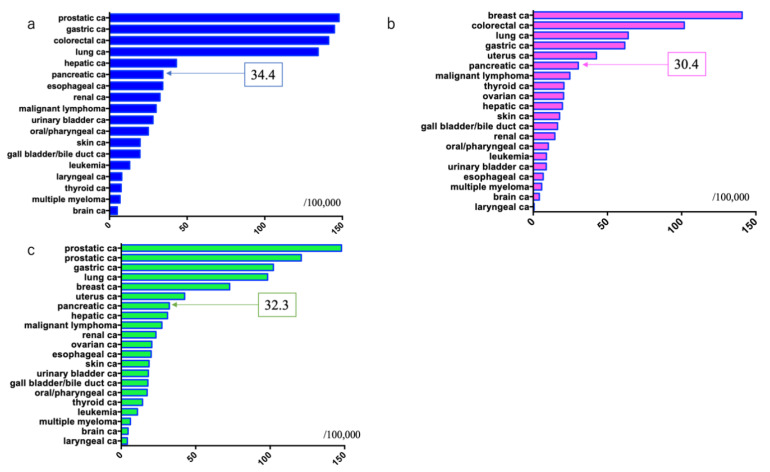
Estimated incidence of pancreatic ductal adenocarcinoma in 2017 in Japan (source: Cancer Registry and Statistics ref. [7]). (**a**) Males; (**b**) Females; (**c**) Both males and females.

**Figure 4 diagnostics-11-00562-f004:**
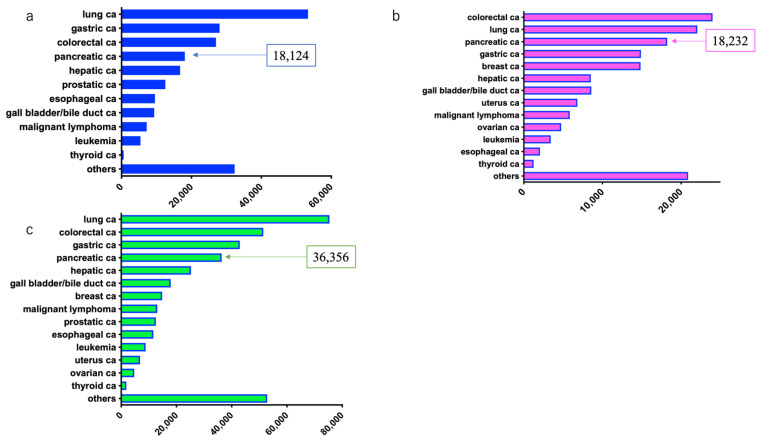
Estimated number of deaths due to pancreatic ductal adenocarcinoma in 2019 in Japan (source: Cancer Registry and Statistics ref. [7]). (**a**) Males; (**b**) Females; (**c**) Both males and females.

**Figure 5 diagnostics-11-00562-f005:**
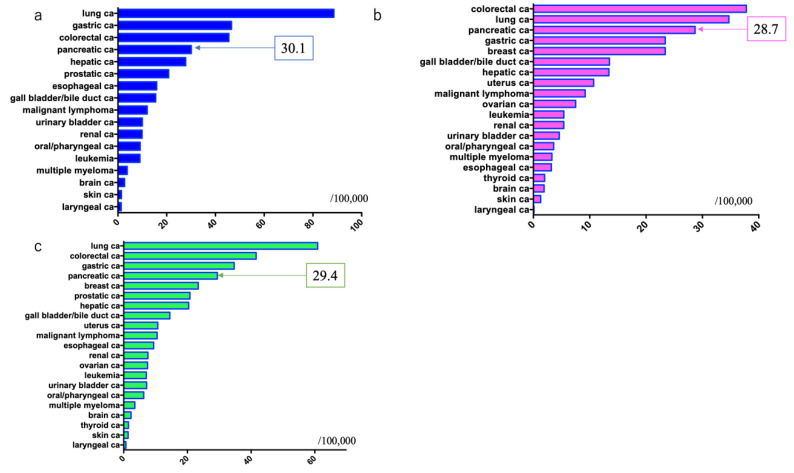
Estimated mortality rate of patients with pancreatic ductal adenocarcinoma in 2019 in Japan (source: Cancer Registry and Statistics ref. [7]). (**a**) Males; (**b**) Females; (**c**) Both males and females.

**Table 1 diagnostics-11-00562-t001:** Hereditary pancreatic cancer syndrome.

Disease	Disease Gene	Hereditary Form	The Risk of PDAC
Hereditary pancreatitis	*PRSS1*	Autosomal dominant	60–87
Hereditary breast and ovarian cancer syndrome	*BRCA1/2*	Autosomal dominant	4.1–5.8
Peutz–Jeghers syndrome	*STK11/LKB1*	Autosomal dominant	132
Familial atypical multiple mole melanoma syndrome	*CDKN2A/p16*	Autosomal dominant	13–22
Hereditary colorectal adenomatous polyposis	*APC*	Autosomal dominant	4.4
Hereditary nonpolyposis colorectal cancer	*hMSH2, hMLH1*	Autosomal dominant	8.6

**Table 2 diagnostics-11-00562-t002:** Risk factors for developing pancreatic ductal adenocarcinoma (PDAC).

	Risk Factors	The Risk of PDAC
Family history	Patients with PDAC in the family	6.79-fold
	Patients with family members with PDAC < 50 years old	9.31-fold
Genetic disorders	Hereditary pancreatitis	67–87-fold
	Hereditary pancreatic cancer syndrome	Refer to Table 1
Complications	Diabetes mellitus	<1 year 5.38-fold, 1–4 years 1.95-fold, 5–9 years 1.49-fold, ≥10 years 1.47-fold
	Obesity	Risk of PDAC onset in males in their 20 s with body mass index ≥ 30 kg/m^2^: 3.5-fold
	Chronic pancreatitis	Within 4 years of diagnosis: 14.6-fold
		≥5 years after diagnosis: 4.8-fold
	Intraductal Papillary Mucinous Neoplasms (IPMNs)	Branch-type IPMN: 15.8–26-fold
Preferences	Smoking	1.68-fold
	Alcohol	1.22-fold
Occupation	Chlorinated hydrocarbon exposure	2.21-fold
Food	Red meat	1.25–1.76-fold

## Data Availability

The data presented in this study are openly available in [GLOBOCAN and Cancer Registry and Statistics] at https://gco.iarc.fr/today (accessed on 28 February 2021) and https://ganjoho.jp/reg_stat/statistics/stat/summary.html (accessed on 28 February 2021), respectively, reference number [2,7,8].

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
