# Peer review of "Pancreatic Ductal Adenocarcinoma: Epidemiology and Risk Factors"

_diagnostics, 2021, doi:10.3390/diagnostics11030562_

Round 1

Reviewer 1 Report

The authors outline the incidence, mortality/morbidity rates of pancreatic cancer worldwide, with a section focused more in depth on the Japan situation.

Considering the relevance of early tumor detection for expected survival, the authors then review known rock factors.

The review is generally well written, clear and has the major strength of listing world-wide numbers, compared to most (if not all) reviews that focus on regional data. The authors do acknowledge significant regional differences and provide hypotheses that could explain those differences. Figures are supportive of the concepts/data proposed.

I have only minor comments and would recommend the publication pending minor edits.

1) Pancreatic cancer is a loose definition that includes a diverse set of malignancies. Pancreatic ductal adenocarcinoma accounts for the majority, yet different forms of pancreatic cancer exhibit major differences in terms of pathology and disease outcome. I would recommend the authors use more caution and would acknowledge disease complexity.

2) Among risk factors, the authors notably omit dietary components. Red meat and goods with high glycemic index have been proposed as significant risk factors (and may also explain regional differences). I acknowledge that experimental data are not compelling and the consensus is not there yet, but those are things worth mentioning, at least in the “others” section.

3) Beyond chronic pancreatitis, also acute pancreatitis has been shown to augment pancreatic cancer risk.

Author Response

Reviewer 1:

The authors outline the incidence, mortality/morbidity rates of pancreatic cancer worldwide, with a section focused more in depth on the Japan situation. Considering the relevance of early tumor detection for expected survival, the authors then review known rock factors. The review is generally well written, clear and has the major strength of listing world-wide numbers, compared to most (if not all) reviews that focus on regional data. The authors do acknowledge significant regional differences and provide hypotheses that could explain those differences. Figures are supportive of the concepts/data proposed. I have only minor comments and would recommend the publication pending minor edits.

Thank you for your kind comments.

  1. Pancreatic cancer is a loose definition that includes a diverse set of malignancies. Pancreatic ductal adenocarcinoma accounts for the majority, yet different forms of pancreatic cancer exhibit major differences in terms of pathology and disease outcome. I would recommend the authors use more caution and would acknowledge disease complexity.

Thank you for this comment. We apologize for the ambiguous description. As you suggested, we changed pancreatic cancer to pancreatic ductal adenocarcinoma (PDAC) to clarify the definition of the term.  

  1. Among risk factors, the authors notably omit dietary components. Red meat and goods with high glycemic index have been proposed as significant risk factors (and may also explain regional differences). I acknowledge that experimental data are not compelling and the consensus is not there yet, but those are things worth mentioning, at least in the “others” section.

Thank you for this comment. As you suggested, we added a description about red meat and goods with high glycemic indices as risk factors for the development of pancreatic ductal adenocarcinoma in the other section on lines 294-298.  

  1. Beyond chronic pancreatitis, also acute pancreatitis has been shown to augment pancreatic cancer risk.

Thank you for this comment. As you suggested, we added a description of acute pancreatitis as a risk factor for the development of pancreatic ductal adenocarcinoma on lines 256-263.  

Reviewer 2 Report

The review article titled "Pancreatic cancer: Epidemiology and risk factors" is a comprehensive documentation of the status of Pancreatic cancer globally and specifically focused on Japan. Before the manuscript can be accepted for publication, some minor changes and modifications are recommended.

Minor comments:

  1. The classification of types of pancreatic cancers need to be mentioned. Specifically Adenocarcinoma, Squamous Cell Carcinoma, Adenosquamous Carcinoma, Colloid Carcinoma, Neuroendocrine Pancreatic Cancer and Benign Precancerous Lesions. Following this then focus on pancreatic ductal adenocarcinoma (PDAC) would be appropriate. This would provide the readers a context of the severity of PDAC.
  2. It is suggested that the title pancreatic ductal adenocarcinoma (PDAC) as an identifier as this review is focused primarily on PDAC
  3. In figures 1 to 5, the font size of the labels needs to be increased. As such, they are too small and difficult to read.
  4. Section 3 ‘Risk Factors’ can be converted to a table for clarity
  5. Table 1 and 2 the top row can be bold fonts for clarity

Author Response

Reviewer 2:

The review article titled "Pancreatic cancer: Epidemiology and risk factors" is a comprehensive documentation of the status of Pancreatic cancer globally and specifically focused on Japan. Before the manuscript can be accepted for publication, some minor changes and modifications are recommended.

Thank you for your kind comments.

  1. The classification of types of pancreatic cancers need to be mentioned. Specifically Adenocarcinoma, Squamous Cell Carcinoma, Adenosquamous Carcinoma, Colloid Carcinoma, Neuroendocrine Pancreatic Cancer and Benign Precancerous Lesions. Following this then focus on pancreatic ductal adenocarcinoma (PDAC) would be appropriate. This would provide the readers a context of the severity of PDAC.

Thank you for this comment. We apologize for the ambiguous description. As both reviewers suggested, we changed pancreatic cancer to pancreatic ductal adenocarcinoma (PDAC) to clarify the definition of the term.  

  1. It is suggested that the title pancreatic ductal adenocarcinoma (PDAC) as an identifier as this review is focused primarily on PDAC

 As you suggested, we changed pancreatic cancer to pancreatic ductal adenocarcinoma in the title to clarify the definition of the term.  

  1. In figures 1 to 5, the font size of the labels needs to be increased. As such, they are too small and difficult to read.

As you suggested, we changed the font size of the labels to be read more easily 

  1. Section 3 ‘Risk Factors’ can be converted to a table for clarity

As you suggested, we revised Table 2 to clarify the risk factors based on section 3 (Risk Factors).

  1. Table 1 and 2 the top row can be bold fonts for clarity

As you suggested, we changed the fonts in the top row in Tables 1 and 2 to be read more easily.